# Biogas Potential Assessment of the Composite Mixture from Duckweed Biomass

**Alexander Chusov, Vladimir Maslikov, Vladimir Badenko \*** **, Viacheslav Zhazhkov, Dmitry Molodtsov and Yuliya Pavlushkina**

Institute of Civil Engineering, Higher School of Hydraulic and Power Engineering Construction, Peter the Great St. Petersburg Polytechnic University, 195251 St. Petersburg, Russia; chusov_an@spbstu.ru (A.C.); maslikov_vi@spbstu.ru (V.M.); zhazhkov@spbstu.ru (V.Z.); molodtsov_dv@spbstu.ru (D.M.); pavlushkina_yu@spbstu.ru (Y.P.)
\* Correspondence: badenko_vl@spbstu.ru

**Abstract:** The article presents the research results of anaerobic digestion processes in bioreactors of composite mixtures based on initial and residual biomass of Lemna minor duckweed and additives: inoculum (manure), food waste, and spent sorbents to determine biogas potential (biogas volume, methane content). Duckweed Lemna minor, which is widespread in freshwater reservoirs, is one of the promising aquatic vegetation species for energy use. Residual biomass is obtained by chemically extracting valuable components from the primary product. The purpose of the research was to evaluate the possibility of the energy potential of residual biomass of Lemna minor to reduce the consumption of fossil fuels and reduce greenhouse gas emissions. This is in line with the International Energy Agency (IEA) scenarios for the reduction of environmental impact. The obtained results confirm the feasibility of using this type of waste for biogas/biomethane production. The recommendations on the optimal composition of the mixture based on the residual biomass of Lemna minor, which will allow for an increase in biogas production, are given. The obtained data can be used in the design of bioreactors.

**Keywords:** biogas potential; methane content; composite mixture; bioreactors; common duckweed; residual biomass; organic waste; circular economy

## 1. Introduction

The technology of anaerobic decomposition of wet biomass with the production of biogas, consisting of 55–60% methane, has been widely used to reduce the consumption of fossil organic fuels and greenhouse gas emissions in many countries around the world (including Russia) [1]. Among the potential energy forms to be derived from biowastes, biogas is of great interest [2]. The processing of biogas into biomethane has become a subject of increased technical interest, as biomethane is expected to be introduced into the national gas grid or used as a vehicle fuel. Moreover, the use of biogas derived from organic matter can support the energy transition [3]. The production and usage of biomethane can provide new opportunities for efficient Circular Economy, but the delay in this could cause significant economic losses [2].

Recently, the use of various aquatic plant organisms as an available raw material for biogas/biofuel production has been recognized as promising [4]. The processing of duckweed has been included in the list of promising pathways for biogas/biofuel production. This property is attributed to its simple harvesting method and high protein or starch content, depending on its species and growing environment [5]. One of the promising species for energy use is duckweed (Lemna minor), the most common representative of higher aquatic plants in freshwater bodies [6,7]. This plant has unique properties: short life cycle, high biological productivity and growth rate (biomass doubles within 2–3 days), wide spreading in various climatic zones of the globe, and undemanding to the quality

of the water environment (used for wastewater treatment) [8,9]. In natural water bodies, the productivity of the duckweed is 0.7–1.0 kg of biomass from 1 m$^2$ of surface. Growing duckweed in artificial conditions is not difficult [10,11].

The biochemical composition of duckweed in terms of the amount of nutrients is not inferior to cereals. The biomass of the duckweed contains about 25.8% protein (twice as much as in cereals), 4.7% fat, and 24.6% fiber (11 times more than cereals). The biomass of the duckweed contains starch, organic nitrogen in the form of protein, and free amino acids, which are useful for biogas production [12,13]. Biomass is used to produce food, medicines, fodder, bioenergy, etc. Lemna minor can be considered as a universal aquatic plant [7,10,14–16]. The widespread use of duckweed will contribute to the challenge of meeting the growing biomass requirements of modern society and will protect the environment by removing excess nutrients and heavy metals from surface water and wastewater [4,17]. These unique characteristics of duckweed make it a technical and medicinal raw material for the extraction of valuable components, such as proteins, lipids, pectins, etc. [18,19].

Publications on the energy potential of duckweed biomass mainly focus on bioethanol production [20–23]. Relatively few research papers concern the possibility of using duckweed for biogas production [24–26]. It should be noted that these reports deal with the initial biomass of Lemna minor, which is not always economically efficient.

The aim of this research is a comparative analysis of the biogas potential of the initial and residual biomass of Lemna minor duckweed, as well as composite mixtures based on residual biomass and additives: inoculum, food waste, and waste sorbents.

## 2. Materials and Methods

### 2.1. Laboratory Research

The initial biomass of the duckweed was collected from the water body in the Leningrad Region in the north-west of the Russian Federation [27]. Residual duckweed biomass is formed after the extraction of pectin substances by hydrolysis. This process takes place in a citric acid solution at pH 1–2 and at 90 °C for 2 h. Then, the plant material is separated from the solution, and the residual biomass is used as a component of the composite mixture for fermentation.

The sorbents are made from carbonized residual biomass of duckweed, with the addition of a chitosan solution as a binder to produce sorbent granules. Granular sorbents were used for the extraction of cadmium, zinc, and copper ions from model solutions [28]. The use of waste sorbents as fermentation additives can solve the problem of their utilization. Fresh cow manure with a moisture content of 82% and an organic carbon content of 92% was used as an inoculum.

The laboratory experiment to assess the biogas potential of prototypes of organic substrates allows a relatively accessible method, with the required accuracy to simulate the processes of biodegradation, considering the influence of various factors (physical, chemical, biological, etc.) [29–31].

### 2.2. Conducting the Experiment

The program of laboratory experiments included the preparation of composite mixtures based on the initial and residual biomass of the duckweed for addition to bioreactors:

- Initial duckweed biomass + inoculum
- Residual duckweed biomass + inoculum
- Residual duckweed biomass + inoculum + food waste
- Residual duckweed biomass + inoculum + food waste + waste sorbent
- Inoculum + waste sorbent
- Inoculum (control sample).

Additives from food waste and waste sorbents were used to assess the impact on anaerobic digestion intensity and biogas production.

For each component of the composite mixture, before loading into the bioreactors, the mass fraction of moisture was determined by the drying method, and the content of organic carbon was determined by the calcination method (Table 1). The moisture content of the samples was determined using an Ohaus MB35 moisture analyzer. The temperature of the drying process was maintained at 105 °C. Drying was carried out automatically to a constant value of the sample mass. Then, the content of organic carbon in the samples was determined by weighing before and after calcining in a PT200 muffle oven. Calcination was carried out at 550 °C for 120 min [32]. The data obtained were used to calculate the mass ratios of the components of composite mixtures based on duckweed.

**Table 1.** Parameters of composite mixtures components.

| Component | Organic Carbon, % of Total Carbon | Moisture, % |
|---|---|---|
| Initial biomass of Lemna minor duckweed | 98.0 | 16.20 |
| Residual biomass of Lemna minor duckweed | 93.0 | 6.44 |
| Food waste | 93.0 | 70.28 |
| Inoculum (Fresh cow manure) | 88.3 | 82.75 |
| Waste sorbent | 87.7 | 7.26 |

Eight bioreactors were prepared for the experiment. Data on the loading of bioreactors in terms of organic carbon in grams are presented in Table 2.

**Table 2.** Contents of composite mixture components (in terms of organic carbon in grams).

| Component | Bioreactor No. | | | | | | | |
|---|---|---|---|---|---|---|---|---|
| | 1 | 2 | 3 | 4 | 5 | 6 | 7 | 8 |
| Initial biomass of Lemna minor duckweed | 4 | | | | | | | |
| Residual biomass of Lemna minor duckweed | | 4 | 4 | 2 | 4 | 2 | | |
| Food waste | | | 4 | 2 | 4 | 2 | | |
| Waste sorbent | | | | | 2 | 4 | | 4 |
| Inoculum (Fresh cow manure) | 4 | 4 | 4 | 2 | 4 | 4 | 4 | 4 |
| Total | 8 | 8 | 12 | 6 | 14 | 12 | 4 | 8 |

*2.3. Laboratory Setup*

To assesses the biogas potential of composite mixtures based on the initial and residual duckweed biomass, a laboratory setup was created. Bioreactors with a volume of 1 L were placed in a thermobox with a constant temperature. The laboratory setup included a discrete mode of anaerobic digestion, in which the bioreactors were loaded with the composite mixture only at the beginning of the process. The duration of the experiments ranged from 35 to 50 days. The laboratory setup layout for testing laboratory samples of the composite mixtures is shown in Figure 1.

Samples of composite mixtures were loaded into bioreactors (Table 2). Then, 600 mL of filtered water was added to these containers. The bioreactors were blown with inert gas to create an anaerobic mode of organic substance decomposition. The bioreactors were connected to Ritter MilliGascounters using gas lines to determine the volume and intensity of biogas emission. The generated biogas was diverted into airtight gas bags with a 3 L capacity. The bioreactors were placed in the thermostatic box, which automatically maintained the constant temperature of 35 °C required for mesophilic fermentation. The bioreactors were equipped with additional gas lines to enable the connection of a gas analyzer for periodic monitoring of the biogas composition. The contents of the bioreactors were mixed daily to prevent floating crust formation on the liquid surface. For remote control of biodegradation processes and monitoring of the experiment in real time, an information-analytical complex was created. This information-analytical complex consisted of a personal computer connected to the control unit of gas meters by wires.

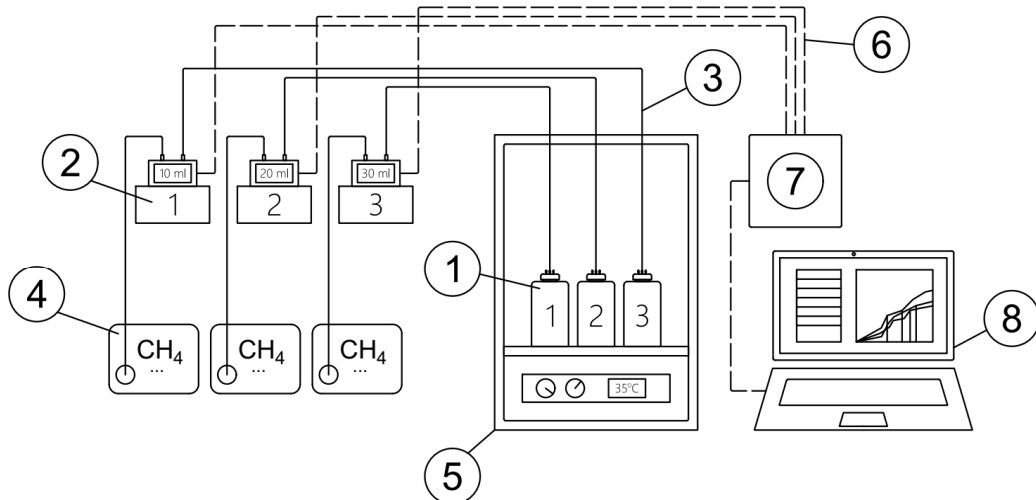

**Figure 1.** Laboratory setup layout: 1—bioreactors, 2—Ritter MilliGascounters, 3—gas lines, 4—gas bags, 5—thermostatic box, 6—wired connection to a personal computer, 7—control unit of the gas meters, 8—personal computer.

In the bioreactors, a discrete mode of anaerobic digestion was carried out, in which the loading of the bioreactors with the samples under study was carried out only once at the beginning of the experiment. The experiment was carried out until the end of biogas emission from the bioreactors.

### 2.4. Process Parameters Monitoring

The hydrogen pH value of the liquid phase was periodically controlled. In the case of low pH value and stoppage of gas emission, a buffer solution of baking soda (10%) was added to bioreactors to increase the pH value to 6.5–7.0.

During the experiment, a scheme for monitoring the component composition of the biogas was used directly in the bioreactor (Figure 2) using the portable gas analyzer GA2000 Plus [32]. Measurements were taken at least once a week. This made it possible to carry out the necessary measurements during the experiment without affecting the component composition of the biogas.

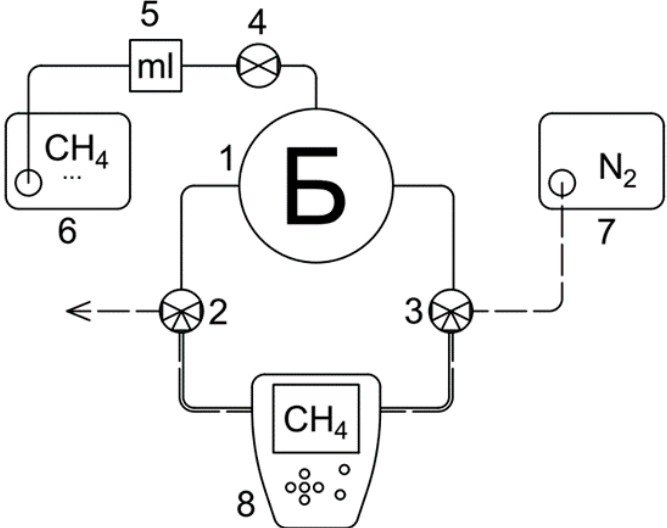

**Figure 2.** The scheme of connecting and measuring the component composition of biogas with the GA2000 Plus gas analyzer: 1—bioreactor; 2,3—three-way valves; 4—two-way valve; 5—gas meter; 6,7—gas bags; 8—gas analyzer.

## 3. Results and Discussion

The biogas-specific emission at biodegradation of the initial and residual biomass of *Lemna minor* duckweed is shown in Figure 3.

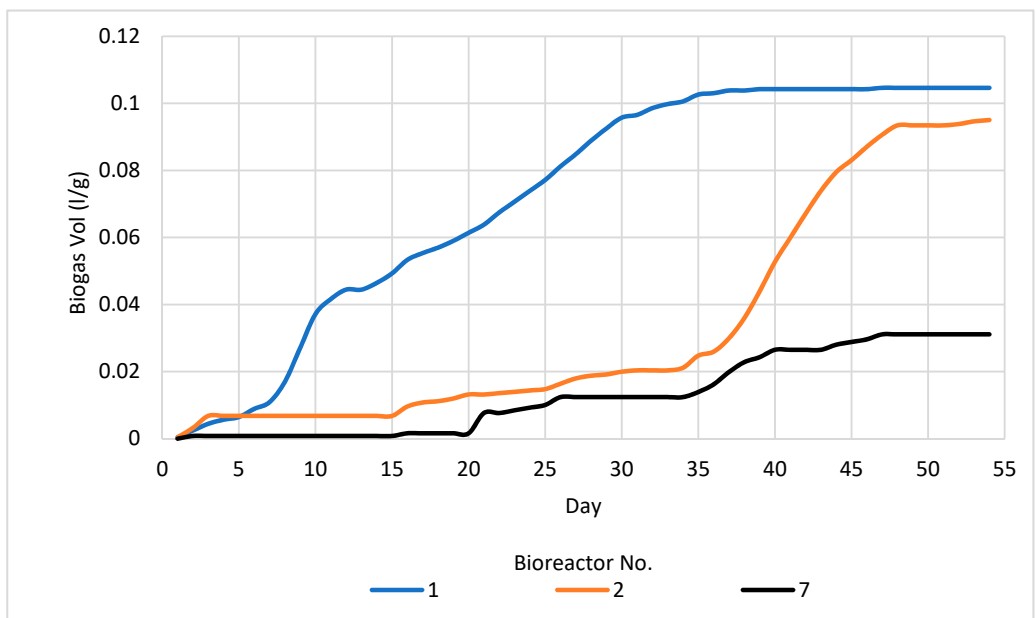

**Figure 3.** Biogas emissions from bioreactors.

The analysis of the presented graph shows that an intensive biogas yield in bioreactor No. 1 (loaded the initial duckweed biomass 50% + the inoculum 50%) was observed from 7 to 35 days during the 50-day experiment; after that, the gas emission practically stopped. The biogas emission on day 35 was 0.103 L/g of organic carbon.

The biogas emission from bioreactor No. 2 (loaded with the residual duckweed biomass 50% + the inoculum 50%) during the same period was four times less, 0.025 L/g of organic carbon, which could be explained by the presence of inhibitory impurities in the sample. The process of intensive biogas production was observed from day 34 to 48 of the experiment. On the 48th day of the experiment, the biogas emission was 0.093 L/g of organic carbon.

There was almost no biogas emission from bioreactor No. 7, loaded only with the inoculum, during the first 20 days. A slight increase in biogas emission was observed from 20 to 48 days. The biogas emission on the 48th day was 0.031 L/g of organic carbon.

The changes in methane concentration in the biogas during fermentation of the initial and residual duckweed biomass are shown in Figure 4.

The analysis of the graphs shows that the stable stage of methanogenesis was observed in bioreactor No. 1 on day 12, characterized by high methane concentration (35%), and a carbon dioxide content of 20.1%. In the following period, the methane concentration increased and reached its maximum on day 35 (45.2%).

The slow growth of methane concentration (from 0.7% to 12%) was observed in bioreactor No. 2 during the first 35 days, with a carbon dioxide content of 15.5–16.9%. Then, the growth rate increased, and by 48 days, the methane concentration reached 45%.

It should be noted that the biodegradation modes of the initial and residual duckweed biomass are very different. In case of the initial biomass, the experiment time was 30–35 days. In case of the residual biomass, the duration of the experiment was increased (up to 50 days), as it took at least 30 days to start a stable biodegradation process.

The active stage of methanogenesis in fermentation of residual biomass started on day 34, when the intensity of the biogas emission and methane content increased.

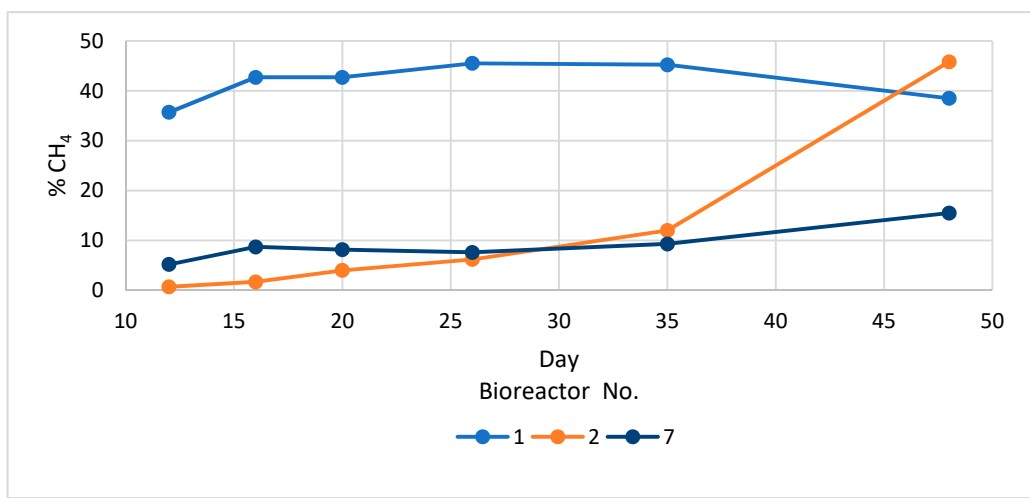

**Figure 4.** Methane concentration in bioreactors.

Figure 5 shows the specific emissions of biogas from bioreactors Nos. 2–6, loaded with composite mixtures based on the residual biomass of the duckweed Lemna minor, and from the control bioreactors Nos. 7–8 (Table 2).

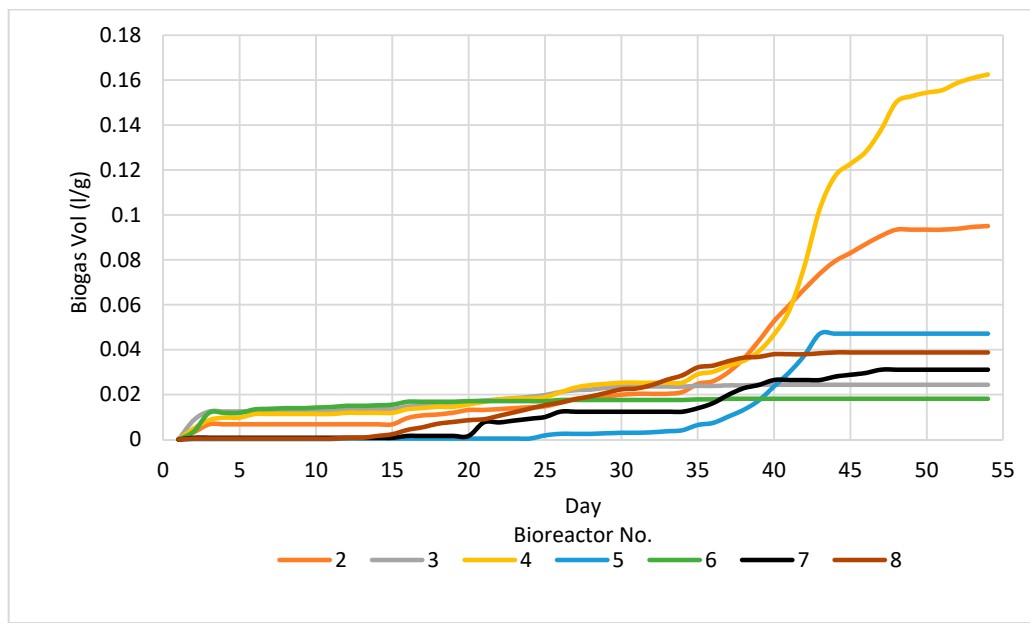

**Figure 5.** The biogas-specific emissions from bioreactors.

Despite the percentage of residual biomass and additives, the biodegradation processes during the first 35 days were slow. On the 48th day of the experiment, the highest specific biogas yield was observed in bioreactor No. 4 (0.16 L/g of organic carbon), which had the following composition mixture: residual duckweed biomass (33.3%), inoculum (33.3%), food waste (33.3%); the total content of organic carbon was 6 g.

The methane concentration changes in bioreactors Nos. 2–8 are shown in Figure 6.

The highest methane concentration for 48 days was also observed in bioreactor No. 4 (49.8%). Therefore, the optimal composition of the compositional mixture, which provided the greatest biogas potential (0.16 L/g of organic carbon, 49.8% of methane) was achieved by the following ratio: residual duckweed biomass (33.3%), inoculum (33.3%), food waste (33.3%).

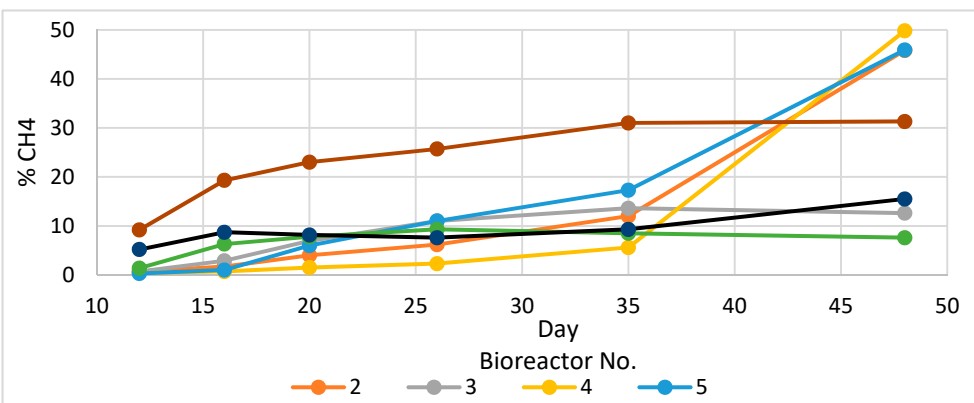

**Figure 6.** Methane concentration in bioreactors.

Directions for future research include the search for methods to accelerate the beginning of the period of intensive biogas emission during the decomposition of composite mixtures based on the residual biomass of Lemna minor. This direction of research is promising not only for the residual duckweed biomass, which is confirmed by the results of studies, for example, as presented in [33] for corn stover waste biomass.

The most challenging task, of course, is the choice of bio-substrates for biogas production. The use of residual biomass for energy purposes most fully complies with the principles of efficient Circular Economy. In this context, it is essential to broaden and deepen the analyses of bio-energy materials that could be used to help to achieve sustainability objectives [34]. Thus, in the future, this work will be also expanded in terms of research on the energy potential of other aquatic plant residual biomass, formed in large volumes after the extraction of valuable components.

## 4. Conclusions

The results obtained allowed us to identify the integral effect of the extraction of valuable components from the primary biomass. Prediction of the biogas potential of the residual biomass of Lemna minor duckweed without experimental studies was difficult because, on the one hand, the chemical treatment of the primary biomass causes partial destruction of a number of cellular structures that are difficult to degrade, and on the other hand, causes an inhibitory effect and a decrease in the content of organic matter in the residual biomass.

It was determined that when the bioreactors were loaded with the same amount (in terms of organic carbon) of the primary and residual biomass of Lemna minor duckweed, the highest specific biogas yield from the residual duckweed biomass was slightly lower, by about 9% (does not exceed the error of similar experiments), than from the primary biomass (with a high methane content of about 50%).

The optimal composition of the mixture was determined based on the residual biomass of the duckweed Lemna minor, which provides the highest specific yield of biogas (0.16 L/g carbon).

Considering the possible scale of Lemna minor duckweed processing and a significant amount of waste generation, the use of residual biomass for biogas production will contribute to sustainable development and compliance with the goals of efficient Circular Economy.

The modes of biodegradation of the primary and residual biomass of the duckweed Lemna minor differed sharply. In the case of primary biomass, the experiment took 30–35 days. When using the residual biomass, the duration of the experiment increased (up to 50 days), because it took at least 30 days to start a stable biodegradation process. This must be considered to optimize the specific loading of bioreactors and calculate their parameters.

**Author Contributions:** Conceptualization, A.C., V.M. and V.B.; methodology, V.M. and D.M.; software, D.M.; validation, V.Z. and Y.P.; formal analysis, A.C., V.M. and V.B.; investigation, V.Z. and Y.P.; writing—original draft preparation, V.M. and D.M; writing—review and editing, A.C. and V.B.; visualization, D.M.; supervision, A.C.; project administration, A.C.; funding acquisition, A.C. and V.B. All authors have read and agreed to the published version of the manuscript.

**Funding:** This research was partially funded by the Ministry of Science and Higher Education of the Russian Federation under the strategic academic leadership program 'Priority 2030' (Agreement 075–15–2021–1333 dated 30 September 2021).

**Institutional Review Board Statement:** Not applicable.

**Informed Consent Statement:** Not applicable.

**Data Availability Statement:** Not applicable.

**Conflicts of Interest:** The authors declare no conflict of interest.

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
