# Peer review of "Biogas Potential Assessment of the Composite Mixture from Duckweed Biomass"

_sustainability, doi:10.3390/su14010351_

Round 1

Reviewer 1 Report

Reviewer’s comments on manuscript sustainability 1506921

General assessment: Topic of the submitted manuscript is both interesting and relevant for Sustainability journal. A thorough revision is necessary. The following points should be addressed:

  • Introduction needs to be enriched as to support the aims of the study
  • Aims of the study need to be clearly stated
  • Materials and Methods is not clear enough
  • Results and the associated Discussion are insufficient and do not allow the manuscript to be accepted as a scientific paper. More results need to be supplied to show the relevance of this study. Discussion is not just a mere description of results, but should contain analytical and comparative aspects – both are absent at present
  • References should be revised / their choice should be justified
  • Language revision is strongly recommended.

Detailed assessment:

Language needs substantial improvement, especially regarding Abstract – proofreading by a native speaker is necessary.

Abstract is too general and lacks qualitative results.

Introduction: Literature survey is too general and the use of lumped references does not help either. Scientific value of the manuscript needs to be highlighted. You should focus on individual publications in the field, extract information relevant to your research as to highlight the actual body of knowledge and the literature gap you intend to fill. This will serve as justification for the goals of your study. At present the goals of your study are not stated which is a major drawback.

Materials and Methods: Please revise this part. Put all important information on system layout and operation in one place. Define the operating conditions clearly.

Results and Discussion part is very scarce on experimental results and the related discussion. Showing specific biogas production and methane content does not make a proper set of results. Or, in other words – this manuscript part is fit as a part of experimental report but does not suffice for an academic paper. Consider showing more detailed experimental data if available (TOC evolution, COD evolution, buffer addition, nitrogen balance etc.). Regarding the discussion, do not only describe the findings depicted in the figures – analyze them deeper, compare with what is known on the digestion of similar substances and mixtures, seek for explanation of observed peak biogas production. Such discussion is one of the key part of a good scientific paper and is a must if this manuscript intends to pass the peer review process.

Conclusions should be revised accordingly when more information is added in the Results and Discussion section.

Formatting: Do use “.” as decimal separator throughout the text.

References: References 7, 18, 22, 29-31 are authored or co-authored by Politaeva N. et al., a researcher from the same institution as the authors of the submitted manuscript. Please, either reduce the share of such citations in the manuscript, or provide a very detailed justification why exactly those papers are important for your manuscript; what unique information they contain. Revise your manuscript accordingly.

Author Response

Dear Reviewer

Thanks for your comments.

All comments were very helpful.

They were fixed and the article really got better.

General assessment: Topic of the submitted manuscript is both interesting and relevant for Sustainability journal. A thorough revision is necessary. The following points should be addressed:

  • Introduction needs to be enriched as to support the aims of the study
  • Aims of the study need to be clearly stated
  • Materials and Methods is not clear enough
  • Results and the associated Discussion are insufficient and do not allow the manuscript to be accepted as a scientific paper. More results need to be supplied to show the relevance of this study. Discussion is not just a mere description of results, but should contain analytical and comparative aspects – both are absent at present
  • References should be revised / their choice should be justified
  • Language revision is strongly recommended.

Detailed assessment:

Language needs substantial improvement, especially regarding Abstract – proofreading by a native speaker is necessary.

Has been improved.

Abstract is too general and lacks qualitative results.

The section has been rewritten.

Introduction: Literature survey is too general and the use of lumped references does not help either. Scientific value of the manuscript needs to be highlighted. You should focus on individual publications in the field, extract information relevant to your research as to highlight the actual body of knowledge and the literature gap you intend to fill. This will serve as justification for the goals of your study. At present the goals of your study are not stated which is a major drawback.

The section has been improved.

Materials and Methods: Please revise this part. Put all important information on system layout and operation in one place. Define the operating conditions clearly.

The section has been improved.

Results and Discussion part is very scarce on experimental results and the related discussion. Showing specific biogas production and methane content does not make a proper set of results. Or, in other words – this manuscript part is fit as a part of experimental report but does not suffice for an academic paper. Consider showing more detailed experimental data if available (TOC evolution, COD evolution, buffer addition, nitrogen balance etc.). Regarding the discussion, do not only describe the findings depicted in the figures – analyze them deeper, compare with what is known on the digestion of similar substances and mixtures, seek for explanation of observed peak biogas production. Such discussion is one of the key part of a good scientific paper and is a must if this manuscript intends to pass the peer review process.

The section has been improved.

Conclusions should be revised accordingly when more information is added in the Results and Discussion section.

The section has been improved.

Formatting: Do use “.” as decimal separator throughout the text.

The issue has been fixed.

References: References 7, 18, 22, 29-31 are authored or co-authored by Politaeva N. et al., a researcher from the same institution as the authors of the submitted manuscript. Please, either reduce the share of such citations in the manuscript, or provide a very detailed justification why exactly those papers are important for your manuscript; what unique information they contain. Revise your manuscript accordingly.

Citation of Politaeva's papers has been reduced.

17 December 2021

Reviewer 2 Report

The Authors presented for evaluation manuscript entitled “ Biogas potential assessment of the composite mixture from duckweed biomass” the subject is actual, however some revisions are needed". Follow up:

  1. The Introduction Section should be extended. The worldwide trend is biomethane add the proper sentence in context to Russia, EU and US. Consider the following reference: https://doi.org/10.1016/j.jece.2021.105944
  2. The multi-citation [1-9] is to wide. Add the proper sentence and cite ONLY most important papers
  3. In English nomenclature, values should be defined with “dot” not comma. 26,4 should be 26.4
  4. In not sure that duckweed can be used worldwide as the biomass. I will consider it in the context of laboratory scale,
  5. Really wrong sentence “Publications on the energy use of the duckweed biomass is mainly devoted to the 50 production of bioethanol [10,15,21–24]” and this citation, revise.
  6. How the organic carbon and moisture were calculated in Table 1? Add the procedure or refernce
  7. Table 2 is entire unclear. Add description, clarify.
  8. The composition of Lab Set-up should be palace under the Figure, NOT in the main text. Revise.
  9. Why the tests was last as many as 50 days? It has an influence on process? If yes, describe.
  10. Authors state “The bioreactors were equipped with additional gas lines enable the connection of a gas analyzer for periodic monitoring of the biogas composition.”  - ADD DETAILS on the equipment – (GE2000) and timing of tests. Add the analytical data, uncerteinity of each test and how many test was repeated.
  11. Figure 2 – what is the UNIT? Only biogas? Why?
  12. ADD AL DATA containing all rectors, not only 1,2 and 7.
  13. Describe the feeding-process in detail
  14.  Figures are CHAOTIC! – please make it readable and clear.
  15. Discussion is SHORT and shallow. Extend.
  16. Conclusion – not all important findings were concluded.
  17. HIGHLIGHT THE NOVELTY ASPECT

Author Response

Dear Reviewer

Thanks for your comments.

All comments were very helpful.

They were fixed and the article really got better.

Comments and Suggestions for Authors

The Authors presented for evaluation manuscript entitled “ Biogas potential assessment of the composite mixture from duckweed biomass” the subject is actual, however some revisions are needed". Follow up:

  1. The Introduction Section should be extended. The worldwide trend is biomethane add the proper sentence in context to Russia, EU and US. Consider the following reference: https://doi.org/10.1016/j.jece.2021.105944
    The section has been improved.
  2. The multi-citation [1-9] is to wide. Add the proper sentence and cite ONLY most important papers
    The issue has been fixed.
  3. In English nomenclature, values should be defined with “dot” not comma. 26,4 should be 26.4
    The issue has been fixed.
  4. In not sure that duckweed can be used worldwide as the biomass. I will consider it in the context of laboratory scale,
    The section has been improved.
  5. Really wrong sentence “Publications on the energy use of the duckweed biomass is mainly devoted to the 50 production of bioethanol [10,15,21–24]” and this citation, revise.
    The issue has been fixed.
  6. How the organic carbon and moisture were calculated in Table 1? Add the procedure or reference
    The section has been improved.
  7. Table 2 is entire unclear. Add description, clarify.
    The section has been improved.
  8. The composition of Lab Set-up should be palace under the Figure, NOT in the main text. Revise.
    The section has been improved.
  9. Why the tests was last as many as 50 days? It has an influence on process? If yes, describe.
    The section has been improved.
  10. Authors state “The bioreactors were equipped with additional gas lines enable the connection of a gas analyzer for periodic monitoring of the biogas composition.”  - ADD DETAILS on the equipment – (GE2000) and timing of tests. Add the analytical data, uncerteinity of each test and how many test was repeated.
    The section has been improved.
  11. Figure 2 – what is the UNIT? Only biogas? Why?
    Yes, only biogas. Since its emission was controlled during the experiment in a continuous automatic mode. The emission of the methane component was determined by calculation by the periodically measured concentration of the component composition of biogas.
  12. ADD AL DATA containing all rectors, not only 1,2 and 7.
    This figure shows the main model mixtures No. 1 and No. 2 (as well as No. 7 inoculum to consider its contribution to biogas emission) to compare the biogas potential of the primary and residual duckweed biomass. Data from other bioreactors are presented below in the article.
  13. Describe the feeding-process in detail
    The section has been improved.
  14.  Figures are CHAOTIC! – please make it readable and clear.
    All figures has been provided with captions and references in the text.
  15. Discussion is SHORT and shallow. Extend.
    The section has been improved.
  16. Conclusion – not all important findings were concluded.
    The section has been improved.
  17. HIGHLIGHT THE NOVELTY ASPECT
    The novelty aspect has been highlighted.

 17 December 2021

Round 2

Reviewer 1 Report

Reviewer’s comments on revised manuscript sustainability 1506921

The authors have dealt with the majority of queries raised in the first review round. As a result the flow, legibility and clarity of the manuscript improved. Still, another revision of the manuscript is necessary as the Results and Discussion chapter – which is very important to make a good academic paper – has not been visibly improved. I really encourage the authors to reconsider that proper discussion does not only describe data and trends from the charts but also compares them with what is known on the research subject in the relevant literature.

Please find attached a bullet point list evaluating the performed revision in more details.

Solved Issues:

Language issues were fixed.

Introduction section has been improved, the use of lumped references is avoided (with an exception in line 73), aims of the study are stated.

Materials and Methods became clearer and easier to read.

The number of references on Politaeva et al. publications were reduced.

Remaining/new issues:

No new/additional results were included in the Results and Discussion part and discussion itself is still meager – in this section the authors did not follow my recommendations and as a result the scientific value of the manuscript remains questionable.

It is very difficult to speak about having identified “optimal” conditions for biogas production given the limited amount of experiments conducted.

Author Response

Answers for Reviewer 1

Dear Reviewer, thank you very much for your comments

The authors have dealt with the majority of queries raised in the first review round. As a result the flow, legibility and clarity of the manuscript improved. Still, another revision of the manuscript is necessary as the Results and Discussion chapter – which is very important to make a good academic paper – has not been visibly improved. I really encourage the authors to reconsider that proper discussion does not only describe data and trends from the charts but also compares them with what is known on the research subject in the relevant literature.

Please find attached a bullet point list evaluating the performed revision in more details.

Solved Issues:

Language issues were fixed.

Introduction section has been improved, the use of lumped references is avoided (with an exception in line 73), aims of the study are stated.

Materials and Methods became clearer and easier to read.

The number of references on Politaeva et al. publications were reduced.

Remaining/new issues:

No new/additional results were included in the Results and Discussion part and discussion itself is still meager – in this section the authors did not follow my recommendations and as a result the scientific value of the manuscript remains questionable.

The "Results and Discussion" section has been improved. The article was enhanced with a good literature.

It is very difficult to speak about having identified “optimal” conditions for biogas production given the limited amount of experiments conducted.

The main purpose of the work was to assess the possibility of energy use of the residual biomass of Lemna Minor duckweed. Additional conclusions in terms of optimal composition were made only for specific mixtures. In the future, the research will be expanded, as stated in the article.

Reviewer 2 Report

Authors prepared revision in accordance with my suggestions. 

Author Response

Dear Reviewer, thank you very much for your comment

Comments and Suggestions for Authors

Authors prepared revision in accordance with my suggestions. 
